# In search of a "vocabulary for recreation": Leisure-time physical activity among humanitarian migrants in regional Australia

Arianne C. Reis●¹‡*, Komla Lokpo²⊙, Matthew Bojanic³⊙, Sandro Sperandei●⁴‡

1 School of Health Sciences, Western Sydney University, Sydney, Australia, 2 Sanctuary Australia Foundation, Coffs Harbour, Australia, 3 School of Medicine, Western Sydney University, Sydney, Australia, 4 Institute of Scientific and Technological Communication & Information in Health, Oswaldo Cruz Foundation (FIOCRUZ), Rio de Janeiro, Brazil

⊙ These authors contributed equally to this work.
‡ These authors are joint senior authors on this work.
* a.reis@westernsydney.edu.au

**Data Availability Statement:** Consent approved for this study and given by all participants was specific and therefore data cannot be shared with other parties or made publicly available. Participants

## Abstract

This study provides an exploration of the meanings of leisure for humanitarian migrants in regional Australia and these meanings' implications for health. It uses mixed-methods to explore leisure-time physical activity participation and day-to-day experiences of leisure and health. A sample representing approximately one third of the Coffs Coast adult humanitarian migrant population completed the survey, as well as 32 individuals who participated in interviews and focus groups. Findings were organised into three themes and explored issues around mental health and time for leisure, cultural differences in experiencing leisure, and the connections with experiences of other disadvantaged groups. We conclude by arguing that leisure needs to be brought to the fore of discussions around service provision and policy making concerned with the wellbeing of the refugee migrant population, particularly focusing on support for the (re)development of a vocabulary for recreation, as a step forward in the journey to healing and belonging.

## Introduction

In 2015, Australia received the third-largest number of re-settled refugees globally, accepting 9,400, or 8.8%, of the 107,100 international resettlement arrivals [1]. With the government's 2015 promise to accept an additional 12,000 Syrian refugees, this number has been increasing steadily, with latest figures indicating close to 22,000 humanitarian arrivals in Australia [2]. This study was initially conceived as an attempt to determine the individual and social factors that influence the patterns of physical activity (PA) and leisure participation among refugees settled in a regional Australian town in order to identify the health implications that could be associated with these patterns. However, it evolved to be an exploration of the meanings of leisure-time physical activity for refugees settled in a new and significantly different social and cultural environment, and how health is implicated in this process. The development of the study in this direction was a consequence of the epistemological position of the authors who

were assured that only the principal researchers would have access to their raw data. This is a vulnerable and small population, and therefore sharing the dataset put privacy at-risk even if individual data are de-identified. Any concerns about the ethical conduct of this research, please contact the Western Sydney University Human Research Ethics Committee through the Research Services office on Tel +61 2 4736 0493 or email humanethics@westernsydney.edu.au. The corresponding author is also available to discuss any matters related to this study.

**Funding:** The authors received no specific funding for this work.

**Competing interests:** The authors have declared that no competing interests exist

engaged in the project, from its inception, as an exploration and conversation with participants who ultimately guided where the project should go and what findings mattered the most. For this reason, it explores, among others, the idea raised by one of our participants of a *vocabulary for recreation*, or lack thereof, among re-settled refugees and how that cuts across, or accurately summarises, the barriers to leisure experienced by this group. As Quirke [3] aptly argues, this is important because "barriers to leisure experienced by recent immigrants and refugees are also barriers to their settlement and integration" (p. 237), and we add that they are also barriers to improving health and wellbeing.

We start by presenting a review of leisure meanings and some challenges associated with established views of leisure for the understanding of the experience of disadvantaged populations, particularly those of refugees. Although our initial study emphasis–and therefore our data–was more specifically focused on leisure-time physical activity, emergent findings from the material collected have broader implications, hence our exploration of leisure meanings more broadly. We then move on to present the methods used in our own study, followed by a presentation and discussion of our findings.

## Leisure meanings

Leisure participation is not an uncomplicated concept. Despite the long history of the study of leisure, famously explored by Aristotle as a pathway to human development in its most holistic form [4], leisure in modern times has been pervasively construed as the antithesis of work [5], being about the 'remaining time', or what people do when they are not fully occupied with the labours of life and the fulfilment of basic human needs [6].

The opposition of leisure to work in modern times is not at all surprising. From the late Middle Ages, leisure had moved from the Greek ideals of human betterment to a synonym for ostentation, luxury and pleasure [7]. The Puritan work ethic that followed in the 1700s that combined family and work responsibilities as services to God and leisure as waste and sin, paved the way for the increase in formal work time during the industrial revolution and the further ostracisation of leisure from "productive", working life [7]. Veblen's [8] seminal work also helped cement much of the perception that "conspicuous" leisure is the privilege of the upper class. In his book, Veblen argued that a significant marker of distinction between the working class and the bourgeoisie in 19th century Britain was the time for leisure and idleness– the working class were not allowed such a privilege. And although non-work time has significantly increased since then, the pervasive notion of leisure as meaningless time seems to have endured to current days.

Another modern meaning of leisure that, in a way, does not move too far from "remaining time" views leisure as the activity one does in their non-work time, such as cultural and recreational activities and pastimes. Implicit in such meaning is the idea that leisure activities are pursued for their own sake, freely and with the sole purpose of enjoyment and fun [9]. Meanings of leisure as time or as activities tend to lead to the quantification and measurement of leisure experiences and pursuits: what activities count and which do not; where leisure starts and when it ends; how much time is used for leisure; how much leisure is good enough and which types lead to better outcomes. Leisure, in this way, is also clearly framed within an industrialised, Western approach to daily living and human pursuits, and implies freedom of choice [10, 11]. However, the (post)modern commodification of leisure pursuits and the power of the superstructure in directing individual choices in capitalist societies challenge assumptions of leisure as being fully freely chosen and directed [12]. Significantly, for minority and disadvantaged groups, including refugee migrants, freedom of choice and positive connotations of leisure experiences can be a distant concept.

In an insightful, original work from the mid-1990s, Russel and Stage [13] problematized the overly positive discourse surrounding leisure and argued that leisure was a burden for the Sudanese women (and men) living in a refugee camp who had an abundance of time but no purpose, apart from surviving the long, endless days in the camp and hoping for a change. At the same time, there are several accounts of the positive contributions leisure can make to refugee resettlement. Leisure can serve important social functions, such as expanding social networks and bonds, improving social cohesion and nurturing deep cultural meanings [14, 15]. Given the mobile existence experienced by refugees, the creation of connections to other individuals, to place and to the new culture in which they have resettled seems urgent and necessary for the establishment of a sense of belonging [16] as well as to supporting a positive sense of wellbeing [17]. Leisure has also been shown to play an important role in activism, social empowerment and the political negotiation of space, rights and immobility [18, 19]. Also significantly, Mathiowetz [20] argued, getting back to Aristotle's writings, that leisure can, and should, support practices of citizenship and is, therefore, a need and a right of all peoples.

It is clear, then, from this short review that leisure's meaning is multi-faceted and that the experience of leisure in the daily life of refugees resettled in a new country is complicated. But we hope it is also clear that leisure is not merely an activity that you pursue in your free time and that it is implicated in social relations that can challenge or conform with current discourses. It is well known that refugee immigrants face serious challenges during resettlement and that poor mental health is a common outcome of this process [21]. However, the connections between the meanings of leisure experienced by refugees and the associated health implications are yet to be comprehensively understood. This study provides a step in this direction by focusing on one aspect of leisure that is frequently engaged in and promoted as a way to improve physical and mental health in Western cultures where a significant proportion of refugee re-settlement occurs: leisure-time physical activity.

## Methods

### Study context

The study was based in Coffs Harbour, a regional town on the Mid North Coast of NSW, Australia. Coffs Harbour hosts a population of around 73,000 people [22] and, at the time of the study, had recently been identified as one of six key regional NSW towns to resettle newly arrived humanitarian migrants in the State. Census data from the year prior to data collection for the present study indicates that Coffs Harbour hosted, at the time, approximately 690 adult re-settled refugees, most of them coming from South-East Asia (33.8%), Sub-Saharan Africa (29.3%), Southern and Central Asia (15.3%) and North Africa and Middle East (13.4%) [22]. However, intake of Syrian and Iraq refugees during 2017 means that refugees from North Africa and Middle East formed an increasing percentage of the local refugee population at the time of the study.

Despite its status as a resettlement area, Coffs Harbour has limited services and opportunities available locally to support social and economic inclusion of resettled refugees beyond government assistance. Statistical data from the 2016 Census confirm this situation: of the 690 adult refugees living in the area, 56.1% were not listed as part of the labour force, 62.3% of them being women. Of those who are employed, almost 55% are in the Agriculture, Forestry and Fishing industry [22]–blueberry farming is locally known as a key employer of newly arrived humanitarian migrants. It is worth noting here that resettled refugees who speak little to no English are eligible to receive up to 510 hours of free English language lessons as part of their resettlement package, and this likely contributes to the high percentage of individuals not listed as a part of the labour force. This means that a significant proportion of resettled refugees

in the region do not have proficient levels of English language and therefore study to gain this proficiency before being able to apply for and be successful in getting a job locally. It is worth noting also that Coffs Harbour displays one of the highest unemployment rates in Australia, with 7.3% of people unemployed, against an average of 6.9% nation-wide and 6.6% in regional NSW [22]. This context makes it very challenging for frequently unskilled humanitarian migrants, with low levels of English proficiency, to find employment.

Within this context, the lead researcher contacted Sanctuary Australia Foundation (from here on Sanctuary), a local charity that has been working in the region for over 30 years, helping thousands of refugees and their families settle in Coffs Harbour and other regions across the country. Sanctuary's knowledge of the local refugee community helped us access the community on their own terms; they also advised us on procedures and greatly supported the recruitment process. For Sanctuary, the opportunity to collect systematic data about the population they serve on a topic that had received little to no attention from government institutions was seen as beneficial to achieving their organisation's aims. In addition, through this collaboration, a resettled refugee was identified and employed as a research assistant and was also instrumental in the development and implementation of the study. He is also a co-author in this paper.

Other local organisations became involved and supported the study during its implementation, helping with recruitment and also with dissemination of findings at the end of the project. A series of presentations were held at local forums to inform key stakeholders of research findings and discuss potential recommendations for action. A project addressing mental health of female refugees through leisure was implemented in the following year as a response to our findings (contact authors for more information).

## Study design

A case study approach was adopted with multi-stage mixed-methods being used for data collection [23]. The methodology utilises positivist epistemology, when focusing on a case study of Coffs Harbour and attempting to objectively assess the levels of engagement in leisure-time physical activity among refugee participants, and constructivist theory-building as a result of one participant's comment that there was no "vocabulary for recreation" available to these refugees. This proved serendipitous in providing a framework upon which the project could be mounted and led the researchers to expand their methodological approach to embrace personal (subjective) understandings of experiences [24, 25].

The study was approved by the first author's institutional Human Research Ethics Committee (Approval Number: H12038), and included full participant consent.

## Data collection

Three interconnected stages of data collection were included in the study. The first stage comprised two focus groups, comprising 7 and 10 individuals respectively, from the target refugee community. The focus groups aimed at providing an initial glimpse into the lived experiences of leisure and PA participation among members of the target community. It was intended to raise initial questions, identify main concerns and provide a knowledge base from which to construct the questionnaires that were subsequently implemented. The focus group schedule of questions is included as a (S1 File) and were based on a broad overview of the relevant literature.

Participant selection was purposeful, with an intention to include a diversity of backgrounds (included Afghanistan, Iran, Myanmar, DR Congo, Ethiopia, Liberia, South Sudan, Syria and Togo), time since settlement (2004–2015), different age groups (19–62), and a balanced number of female (n = 7) and male (n = 10) participants to provide richness to the

discussions. Given this phase of the study was exploratory in nature and intended to inform the development of the survey, no data saturation was pursued and sample size was determined based on achieving a diverse representation of the local refugee population and ideal size of focus groups–research on focus groups has recommended that a minimum of four and a maximum of twelve participants be included in every one session to avoid silent voices and confusion, but still provide enough quorum to entice useful and productive discussions [26].

The second stage of the study focused on quantitative data collection. Two recruitment procedures were utilized in this stage. First, respondent-driven sampling (RDS) was used to recruit participants [27]. In this case, 5 individuals were identified by our local partner and invited to participate in the study, being the first 'seeds' in the recruitment process. Each seed was given two 'invitation cards' to give out to other members of the community who fitted the eligibility criteria and, through this process, we achieved control of the social network structure upon which this method is dependent. One hundred and thirty-nine participants were surveyed through this process (Survey 1), which came to a stagnant phase after 12 weeks. Given the desire to reach a larger sample of the population, the survey process moved then to a convenience sample (Survey 2) and, in 11 weeks, 93 individuals were reached.

The surveys were divided into three sections. Section 1 comprised questions from the International Physical Activity Questionnaire (IPAQ), a validated and reliable tool to assess levels of PA among the general population [28], as well as a series of questions related to factors preventing participation, constructed based on the literature and on issues identified by participants in the two focus groups. Section 2 consisted of socio-demographic questions as well as questions about the participants' health. Here, relevant questions (particularly based on issues identified during focus groups) were selected from national health surveys as well as questions from the Kessler Psychological Distress Scale (K10), a reliable and validated tool that has been extensively used in cross-cultural research [29] to identify symptoms of psychological distress. The final section aimed to collect information about the participants' personal networks, following guidelines for RDS studies [30]. In addition, a space was provided at the end of the survey for participants' comments. The survey is included as a (S2 File).

Surveys were completed in English only. Interpreters from the local community were available during data collection (either onsite or via telephone) to support those who needed assistance in completing the questionnaire.

The final data collection phase consisted of in-depth, semi-structured interviews with members of the Coffs Coast refugee community. At this stage, issues identified in the findings of the surveys were explored with individual members to uncover deeper meanings and the reasons why some of the behaviours and perceptions are present within their community. The interview schedule of questions is included as a (S3 File).

In total, fifteen individuals were interviewed. Interviews were finalised once data saturation was reached [31]. Similar to focus groups' participants, selection was purposeful, with the intention to include a diversity of backgrounds (included Afghanistan, Burkina Faso, Burundi, DR Congo, Ethiopia, Liberia, Myanmar, Syria and Togo), time since settlement (2004–2016), different age groups (19–58), and a balanced number of female (n = 7) and male (n = 8) participants.

Eligibility criteria for participation in all stages of the study was broad and included any adult (i.e. 18 yrs or older) who self-identified as a resettled refugee member of the Coffs Coast community.

## Data analysis

Data analysis was fully integrated and occurred in various stages given the amount and complexity of the data. Below we explain the distinct processes for each type of data (i.e.

quantitative and qualitative), but it is important to note that they did not occur in isolation. For instance, initial focus group exploratory analysis occurred prior to the development of the survey, and initial explanatory analysis of the surveys was conducted prior to interviews. However, analyses were re-visited and further theoretically explored once all data had been collected. This process is not uncommon in multistage mixed methods studies and provides a powerful strategy to achieve full integration of the material collected [23].

**Qualitative data analysis.** A five-step analysis, following Braun and Clarke [32] thematic analysis procedure, was utilised. The transcribed material from focus groups and interviews was first organised into nodes, following an inductive process, with no preconceived classifications. Subsequently, nodes were grouped into broader themes, which were then reviewed for inconsistencies, overlaps and redundancies. The final step involved the naming of themes and identification of phrases and rich quotations that strongly encapsulated these themes. The quotations were then used in the reporting phase in order to enrich the discussion of the results generated. The software for qualitative data analysis NVivo 10 was used as a data management programme to assist in the organisation of the material and in the identification of themes.

To ensure trustworthiness, triangulation was conducted through extensive review and discussion of findings between three members of the research team, one of whom a member of the refugee community himself and therefore someone with a lived experience of being a refugee. Also, the various sources of data, including quantitative data, provided a complex but intricate and well-connected picture of the findings that supported the triangulation process. In addition, participant quotes are provided here to enrich the description of participants' experiences and to provide readers with the opportunity to scrutinize the researchers' interpretations [24]. Although member checking was not possible, particularly due to low levels of literacy of several focus group and interview participants, a member of the research team who is a refugee community leader confirmed that the findings resonate with the experience of many in his community.

**Quantitative data analysis.** All survey responses were entered into an Excel spreadsheet for analysis. Data were entered by two research assistants independently, and then cross-checked to identify any typing errors. Identified errors were subsequently corrected and a clean spreadsheet was used for analysis using R software, version 3.4.1.

Given that only part of the sample was collected through the RDS method, RDS estimators were not utilized, with the entire sample being combined and analysed as a convenience sample. Participants' degree distribution in the RDS sample suggests that this strategy did not have a major deleterious impact on results and, therefore, the validity of findings is not compromised [30].

The first step in the analysis of the surveys was to calculate the total PA performed by each individual, following IPAQ Scoring Protocol [33]. Subsequently, participants were classified according to their level of PA in a "high", "moderate", or "low" level category. A second classification was performed, with classes "high" and "moderate" being grouped as "Sufficient Level of PA" (individuals who met the World Health Organisation's [34] minimum recommended weekly PA) and "Insufficient Level of PA" (those who did not reach WHO's minimum recommendation).

A univariate descriptive analysis was performed using central tendencies and dispersion measures. Subsequently, a series of univariate linear models was used to select the most prominent variables. Due to sample-size restrictions, the fifteen variables with the best relationship with level of PA were entered in a multivariate logistic model to assess the impact of different variables on the quantity of PA reported, measured in METs/wk. The modelling process started with a full model; then, variables were dropped one by one and the model fitted again, until the final model, containing only significant variables, was reached [35].

## Results

A total of 232 surveys were fully completed and used in the analysis. According to the Australian Bureau of Statistics 2016 Census [22], this sample represents approximately one third of the entire Coffs Coast adult humanitarian migrant population in 2016 (≈690), and presents a distribution very similar to that described by the Australian Bureau of Statistics [22] for the refugee population in the region in regards to region of origin, religion, sex, age, and year of arrival in Australia. The most marked difference in comparison to Census data comes from the second stage of quantitative data collection, which captured the recent (i.e. post-Census) influx of Syrian refugees. Nonetheless, this difference is welcome, as it makes the sample more up-to-date in relation to the current refugee population in the region.

The survey sample comprised 57% female and 43% male, coming from a wide variety of countries, including the Democratic Republic of Congo (15.6%), Syria (15.2%), Burma/Myanmar (12.6%), Iraq (12.6%), Afghanistan (10%), Ethiopia (7.4%), Togo (6,1%) and Liberia (5.2%). The majority of participants were young adults aged between 18 and 29 years (34.4%), followed by those aged between 30 and 39 (25.2%) and 40 to 49 (22.6%) years. The majority was married (51.7%) and close to half of respondents had only been in Australia for up to 2 years (47.4%), although a considerable group had been living in Australia for more than 8 years (14.6%). Most participants had no previous schooling (28.9%), although the majority were currently studying (51.9%), most of whom would be completing their Adult Migrant English Program or some vocational training offered locally. Most participants who were not studying were unemployed (13.9%), working casually (9.5%) or part-time (9.1%). The average number of people living in the same household was 4.6 and the average number of children 2.6. More details regarding the demographic profile of participants is presented in the S1 Table.

In regards to health indicators, participants' reports were generally positive, with almost the entire sample being comprised of non-smokers (95.3%) and indicating they had less than one alcoholic drink per week (92.2%). The vast majority (73.3%) also indicated they had never been diagnosed with common non-communicable diseases.

Results from the SF-36 and K10, however, present concerning numbers. Female respondents were particularly affected in both cases, presenting moderate to severe limitation scores in the SF-36 survey (41.2%) and moderate/severe levels of psychological distress according to K10 results (23.5%), while 34% of male participants presented moderate/severe limitation scores in the SF-36 survey and 13% presented moderate/severe levels of psychological distress. More details regarding health indicators of the sample is provided in the S2 Table.

In the following sub-sections, further quantitative results are presented and discussed together with qualitative ones in order to facilitate integration of the knowledge gained from all phases of the study, where the rather "dry" numerical data can be nuanced and further explored for meanings and context. The themes from the qualitative phase of the study guide the organisation of the material, as they provide further depth to the results found in the quantitative phase of the study. They are divided into three: "The Whole Life is Overwhelming", where we explore issues around mental health and time for leisure, particularly leisure-time physical activity; "It Makes You Lazy", where cultural differences in experiencing leisure-time physical activity are discussed; and "A Similar Story", where connections with experiences of other disadvantaged groups are highlighted.

### "The whole life is overwhelming"

Interview and focus group participants recurrently reported mental health issues, not only experienced by themselves since arriving but also by other family members, particularly women who are more frequently carers in the household, as the quote below exemplifies:

*Yes, the trauma in the life, I personally feel ok here, children are ok. My wife has some mental health issues, she is not very much satisfied with how much she is being understood so, things don't change and it has problems where she has to live on medications and, she's not happy with herself (Male, 49, from Ethiopia)*

Participating women frequently reported feelings of social isolation and described how their busy lives revolved solely around caring for young children and partners. In this context, participation in PA was seen as superfluous and a privilege they could not afford, both in terms of finance, time and also motivation. The quotes below provide a glimpse into some of the challenges experienced by refugees who, understandably, struggle to even allow themselves the opportunity to engage in leisure:

*You know, to have an enjoyment time, a man has to be in a peaceful manner or in very joyful manner. But if the man is not in a joyful manner, I don't think you will think of being happy and do activities and all things like that. [. . .] I come from South Sudan, and South Sudan up to now is still in a big struggle and things, and we are connecting. [. . .] So what's happing there affects me here. And hits me, and I don't have time for myself really to enjoy anything. (Male, 47, from South Sudan)*

*I don't have a good head to do any activity. Before I was a soccer player in my country. But when I have become refugee, a refugee life! I don't wish this life to anybody. [. . .] You lost everything. Every day no good life for refugee. [. . .] I am in Australia. I thank God to provide such an opportunity to me and my family. But there is no job to do. We also have language barrier. [. . .] This is a very big problem for me. I don't have anything to think, doing any activities like sports (Male, 62, from Togo)*

Accounts from interview and focus group participants highlight some of the major struggles faced by refugees settled in the Coffs Coast that also have particularly intense impacts on the physical and mental health of participants. A female participant, from Sierra Leone, described her struggles having to manage a busy life looking after children, studying to improve her English and work prospects, on top of a life of displacement and worries:

*A lot of it has to do with mind-set. When we look to responsibilities, although we are here, some of us are carrying extra baggage because we are here and you are also thinking of home. These are things that are impact on so many things. Because for those who are here, you have to think about the kids. Because if the kids are at school the mothers would relax a little bit. But then, if there are no jobs for them, they have to reminder they have to go to Centrelink, I have to do this, I have to do that [. . .] And when you have to take care of kids and think of recreation. Some don't even have the vocabulary for recreation. [. . .] It's not that they don't know how to do it. It's like they are overcrowded and there are so many issues that you need to take care of before you can start (Female, 47, from Sierra Leone)*

The "vocabulary for recreation" mentioned by this female refugee is fundamental for understanding the lack of participation in leisure activities found among humanitarian migrants settled in the Coffs Coast and will be explored further in our discussions.

Some participants talked extensively about persistent pain and suffering, particularly because of how they continue to deal with past experiences vicariously through news of war and crime in their countries of origin, or countries where they lived for years as asylum seekers:

*The other thing is, the whole life is overwhelming. Even when you have plenty of time you don't use it. You just wonder. Time just goes by and you don't do much. And a lot of other things that are affecting us and [male from South Sudan] had mentioned it, what happened there affects us. There is war in South Sudan and he is listening to the news every day. And there is a big problem in Ethiopia. People are dying and everything. I am listening to that. And it's really painful. So every time I go home in the evening all I do is listening to the news. That paralyzes me. I can't do anything. I can't think of play. Any leisure. (Male, 49, from Ethiopia)*

The "vocabulary for recreation" becomes damaged and ever harder to re-learn, a point to which we will return in our discussions.

### "It makes you lazy"

Results from the IPAQ survey show high levels of PA engagement (58.6%) among Coffs Coast refugee migrants, with only a small proportion of the sample indicating achieving low levels of PA (9.1%). When combined into sufficient and insufficient levels of PA, the absolute majority (90.1%) reaches WHO recommendations. However, when we distinguish leisure-time and transport-related PA from occupational and household PA (Fig 1), it becomes apparent that a significant proportion of the PA happens outside the realm of leisure. Findings from the qualitative phase of this study corroborate these results.

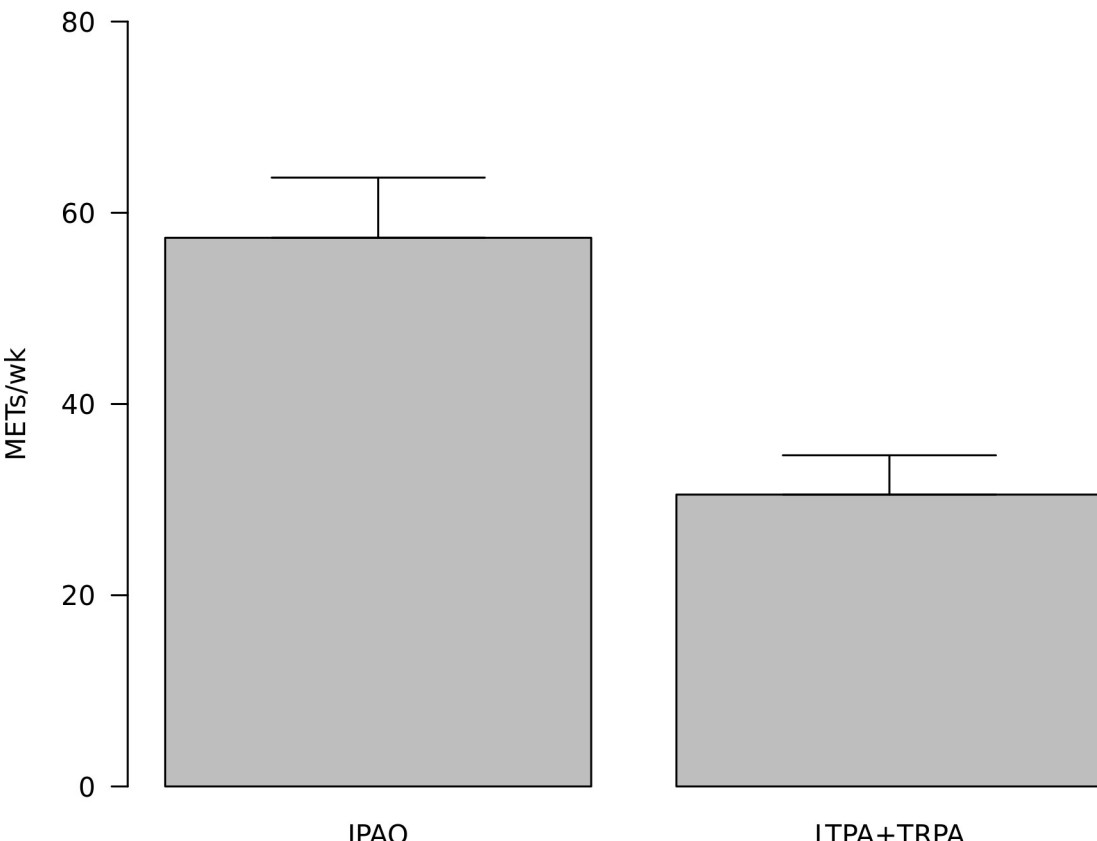

**Fig 1. Full IPAQ results versus leisure-time physical activity plus transport-related physical activity only.**

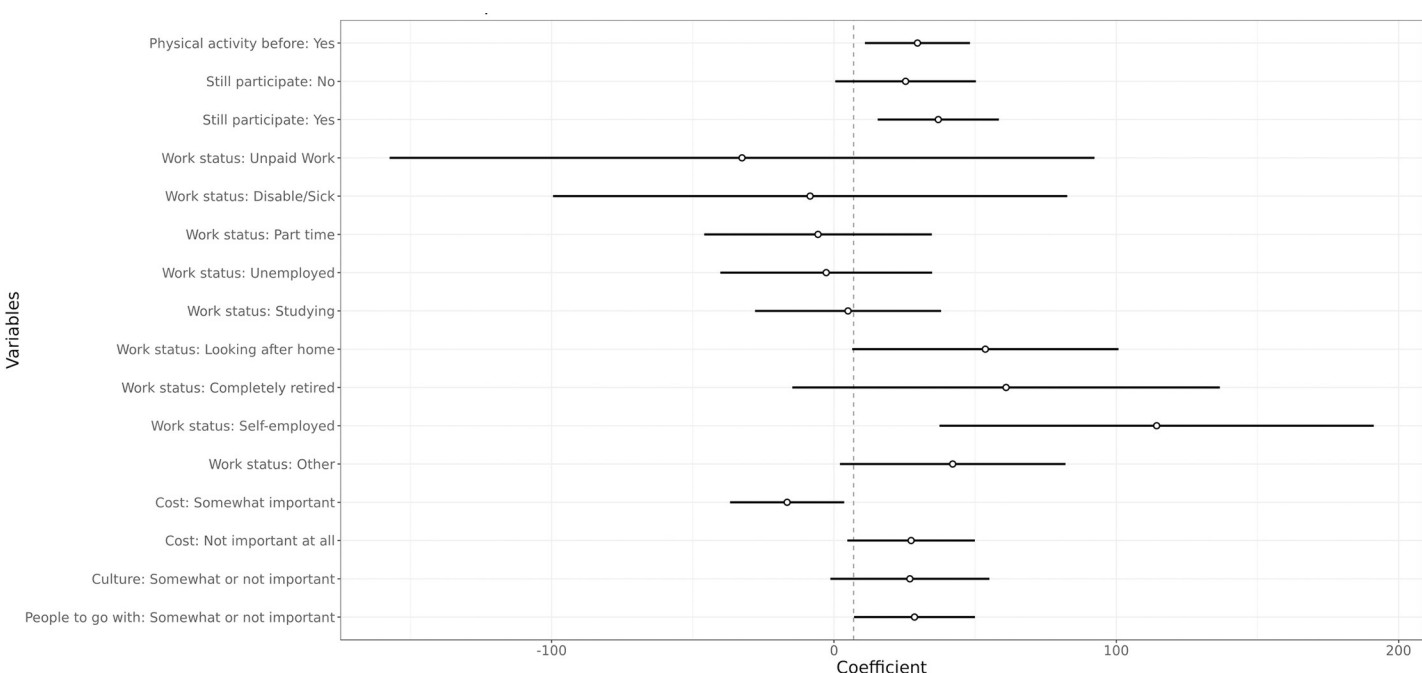

**Fig 2. Final model—effect of sample characteristics on the estimated amount of METs/wk.** Variables associated with physical activity participation. The intercept is composed of individuals who: did not start any new physical activity since arriving in Australia; did not engage in physical activity in their original country; declared a full-time job; declared cost as a very important barrier to physical activity; declared culture as a very important barrier to physical activity; declared not having people to go with as a very important barrier to physical activity.

The final model derived from the process of determining the correlates of PA participation among this population is presented in Fig 2. As expected, having started a new PA since arriving in Australia was associated with higher energy expenditure. These individuals, on average, expend three to four times the energy of those who did not engage in new activities since arriving. Also, those who used to engage in PA in their home country, although not engaging in PA any more, were more physically active than those who never engaged in PA at all. And, again as expected, those who are still engaging in the same activities are the most active ones. Self-employed, those who are looking after the home and those who described their status as "other" are more physically active than full-time employed individuals. Those with unpaid work were the least active, although a great variability was observed.

It is interesting to note that the majority of participants indicated that they engaged in leisure-time PA before arriving in Australia (65.5%), but this number dropped when asked if they continued their engagement with that activity after arrival (41.6%), which suggests the challenges associated with resettlement have an impact on engagement. However, as we can see in Fig 3, there is some evidence to suggest that the learning of a new vocabulary for recreation is happening among at least part of this group, with considerable numbers starting a new activity since arriving in Australia.

Exploring this point in more depth in interviews and focus groups, it became apparent that the positive change in leisure-time PA participation among some humanitarian migrants may be due to a new lifestyle that allows for more freedom of movement and choice:

*Interviewer*: *Do you think what you do here [in terms of leisure and PA] is different from what you used to do in your home country?*

*All participants: Yes, absolutely.*

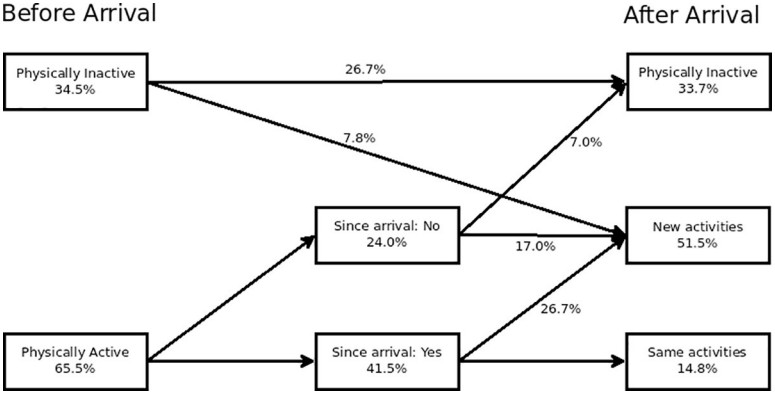

**Fig 3. Activity flow.**

*Male, 19, from Myanmar: there wasn't time there. [. . .]*

*Female, 24, from Afghanistan: Here, there are more opportunities. Not like any distraction or any rules, any particular rules. Anyone can do whatever they want.*

Other participants also shared this feeling:

*I can't ride a bike at my home country, but I can ride here. (Female, 21, from Afghanistan)*

When I was in Iran, I used to walk. My physical walk. But the thing was like there you are forced to walk. It was not like happiness, it was more sadness. . .But here I work, I like my job, but I do sport for fun. And that makes me more happy. I choose what you want to do. That makes me more happy. [. . .] it's a good thing of the freedom we have, to choose what you want to do. (Male, 21 from Iran)

It is worth noting that several participants highlighted how life in their home countries or refugee camps invariably involved more physical engagement than in Australia and that leisure-time PA is a (Western) concept that may not be applicable to other cultures.

*Where I came from, because of the life style, people are physically active because they have to walk. So you don't do physical activities for the sake of doing physical activities. You're doing your job and at the same time you are doing physical activities. (Male, 49, from Ethiopia)*

*So, the difference here is when you come to physical activities, [. . .] there we do because we do what we are supposed to do and at the same time we are physical active. Here, you are not so active, you need to have your own plan to do physical activities intentionally. You need to go for a walk. If you don't want to go for a walk, you need to do it because you want to be physically active and they also tell you there are problems if you are not physical active. Even the doctor tells you if you go or to say: Do you do some physical activities? And when you say no. That's the danger. That's the problem. (Male, 70, from Liberia)*

Along these lines, participants in focus groups and interviews emphasised how the change in lifestyle impacted on their engagement in PA:

*The first [. . .] four years after we arrived here I did not start driving and I was really walking, carrying goods from the shop and up the hills. [. . .] Once I got a driver's licence, now things*

*changed. Sometime when I stopped the car somewhere far away like I say let's go to the shop and they said ah no it's too far away. Even the mum. Yeah, it's hard. It makes you lazy. (Male, 49, from Ethiopia)*

## A similar story

There are certainly many challenges associated with learning the vocabulary for leisure when refugees settle in a new culture after forced migration. However, some of these challenges are not too dissimilar to those experienced by other marginalised groups in our (Western) societies. When looking at the main barriers for participation in PA, cost, language barriers, time and lack of knowledge of where to go feature highest on the list (Fig 4); however, only cost showed a significant relationship with levels of PA engagement (Fig 2).

Interestingly, culture did not feature highly in the list of barriers, but did present a significant correlation with actual levels of PA participation (Fig 2). Those who indicated that culture is a somewhat important or not at all important barrier for participation engage in more PA

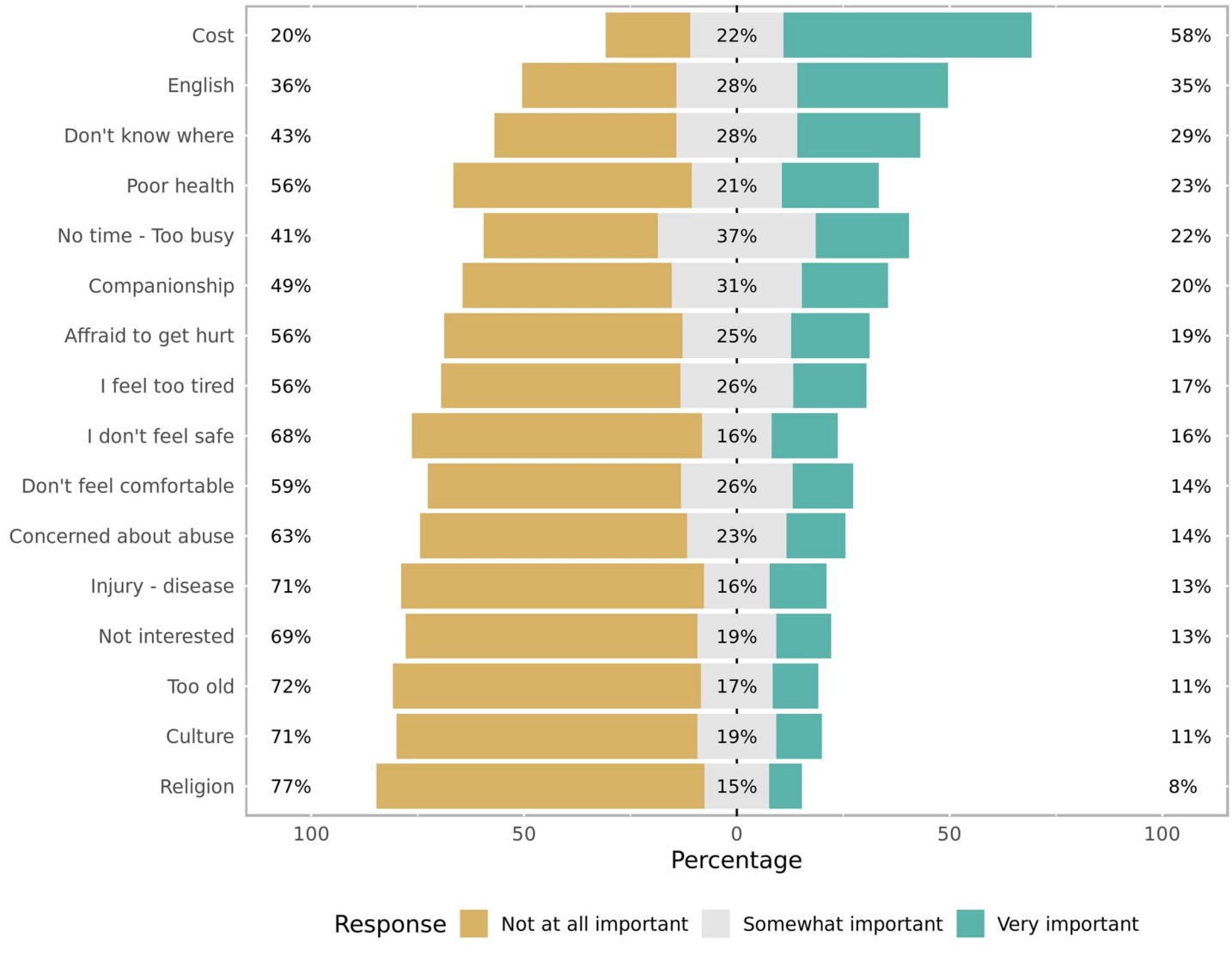

**Fig 4. Barriers to participation.**

than those who feel that culture is a very important barrier for participation. Therefore, although culture does not feature as a frequent barrier for PA participation, it seems that it is a strong barrier for those who feel it is important. Our qualitative results point in the same direction:

> *For example, for us, Afghan families, like girls, she's not allowed in our country, they cannot do many things. But here they are more open. But still some families "lock" the girls, they don't let them to do what they want. For example, my sister she wanted to go to gym, she wanted to go to fitness, go work out, go for running or swimming. For me and my family we allow it, anything she wants to do. I have just signed up her to gym. But I know some other families, even though they are here, they know they have that freedom, but they still do that to their families. They do that to themselves. This is the hardest thing for refugees. They don't change too fast. (Male, 20, from Afghanistan)*

> *Because in my culture, women have different rules and boys have different laws. We are not equal. We're all living in Australia but we are not equal. [. . .] In my culture, if boys do anything that come from any other culture, it's ok. But if girls do, [. . .] it becomes a very big issue. So I feel like in my culture we are not equal, we are not in equal right. (Female, 31, from Myanmar)*

However, our quantitative study did not find gender-based differences in participation, suggesting that other factors not captured by our survey might be at play in this community and we are not aware.

Companionship also seems to play an important role: those who indicate that companionship is not an important, or is only a somewhat important, barrier for participation engage in less PA than those who indicate that companionship is very important. One of the interview participants summarised well the importance of companionship:

> *Doing it with other people is really good because you're always learning new stuff. But doing it by your own sometime you don't feel motivated, you feel lazy. But if you have someone with you, he will push you. I always want someone to come with me. (Male, 19, from Burkina Faso)*

## Discussion

This article explored the different meanings and practices of leisure found in the experiences of day-to-day living of refugee migrants resettled in a regional Australian town. In particular, we were interested in exploring the implications for health, and used leisure-time PA as a main marker of engagement, although not limiting to that form of practice.

In general, participants' reports regarding their overall health was positive. Health indicators, such as smoking habits, alcohol consumption and diagnostic of common non-communicable diseases, were largely positive. However, these results need to be seen with caution as lack of diagnosis, in particular, may be caused by lack of access to medical care rather than the absence of the condition. Previous research has found a high prevalence of undiagnosed conditions among refugees arriving to Australia, including communicable and non-communicable diseases [36], as well as significant barriers to accessing health care after resettlement in Australia [37, 38].

Survey participants also reported high levels of engagement in PA, which can be considered a positive health indicator. However, most of the engagement reported comes from household and work-related duties, which research has repeatedly shown to be less conducive to positive

health outcomes [39, 40]. Conversely, findings from the qualitative phase of the study highlighted the lack of engagement in leisure-time PA among refugees, a finding not dissimilar to those of studies with other vulnerable communities conducted elsewhere [41]. LTPA has not only been shown to be more effective in improving physical health outcomes [40] but engagement in leisure, more generally, has long been recognised as a human right and as a crucial part of what it means to be human. From Aristotle in ancient Greece to the Universal Declaration of Human Rights in 1948 to many contemporary studies confirming the objective and subjective benefits of leisure to wellbeing [42], the literature is full of accounts of the power of leisure engagement to personal development [43, 44] as well as to physical and emotional health [45, 46]. However, despite this "common" knowledge, leisure and recreation are still not equitably framed in health policy discourse and practice [47], and individuals and communities, as a consequence, are not appropriately equipped and empowered to "decipher" or construct a vocabulary for recreation, as our participation aptly describes it. In the case of refugee migrants, this issue is exacerbated by the frequently continuing experiences of trauma even after resettlement [48] and the acute challenges faced when settling in a new culture [49].

One of the emergent themes of our study was the idea that "the whole life is overwhelming" for refugee migrants resettled in the Coffs Coast. Our participants repeatedly reported experiencing poor mental health; not only themselves but also other family members, particularly women who are frequently carers. Results from K10 questions revealed concerning levels of psychological distress, reaching almost one in five respondents. Women were particularly vulnerable, presenting significantly higher rates than men (23.5% and 13% respectively). In Australia, K10 rates for 2014–15 were at 14.5% for women and 11.3% for men [50], showing a considerable disparity between Coffs Coast refugees and the general Australian population.

There were several accounts of experience of social isolation after settlement in Australia; again, particularly among women in our sample. This social isolation was frequently the result of caring responsibilities for a large number of children while husbands were at work, but also due to lack of familiarity with the culture, lack of fluency in English, and disconnection from place given the significant connections with remaining family members in war zones or refugee camps elsewhere. Although several male participants reported similar feelings, it was evident that females were more home-bound, making them more vulnerable to experiencing isolation, which frequently led to severe psychological distress. In this context, participation in leisure activities was seen as superfluous and a privilege they could not afford, both in terms of finance, time and also motivation. Hartley and colleagues [51] found a similar result, where the stress arising from constant worry about the future in Australia meant that refugees in their study reported having no energy to consider partaking in PA. Even though research has repeatedly shown the potential benefits of leisure-time PA for refugees, and war, trauma and torture survivors [52–54], mental health and other resettlement challenges often lead to disengagement in sport and exercise [55].

Most refugee migrants in Australia have, at some stage, experienced traumatic events, such as being witness to or a victim of extreme violence, human rights abuses, persecution, and loss of identity, culture and of family members, all of which have serious psychological impacts on individuals and their families [21], and our participants were no different. Although poor mental health has been repeatedly reported in the literature [56, 57], the impacts on leisure participation, and leisure-time PA in particular, has been less explored.

The concept of a vocabulary for recreation, as raised by one of our participants, provides a framework for understanding leisure engagement among refugees in this study. There are certainly many examples of the healing and restorative power of leisure engagement for refugees and other groups who have suffered severe trauma [52, 58], but what our participants highlight is the need to take into account the re-learning process that it entails, as well as the need for an

acceptance of the right to leisure [59]. As the quotes presented in the results section repeatedly demonstrate, there is a strong view of leisure as an indulgence, something that can only be enjoyed by those who have the luxury of having the means and free time to do it, just like Veblen argued more than a century ago [8]. Such a position may be recurrent in their narratives because it is equally entrenched in public discourse.

Our results present further insights into the meanings of leisure for refugee migrants. Our second emergent theme, "It Makes You Lazy", refers to the cultural differences in leisure engagement, or the different meanings different cultures and experiences create of leisure. The majority of participants indicated that they engaged in leisure-time PA before arriving in Australia, but this number dropped after arrival, suggesting that the challenges associated with resettlement does indeed have an impact on engagement; a finding that is not surprising. Other studies have found an association between culture and cultural norms and PA participation among various migrant communities, suggesting that female participation, in particular, can be seen as culturally inappropriate, not feminine or as neglecting family duties [60–62]. Nevertheless, learning of a new vocabulary for recreation is possible and our findings suggest this is happening among at least part of this group, with considerable numbers starting a new activity since arriving in Australia. Their accounts associated their renewed leisure engagement with a (Western) lifestyle that "allows" for more freedom of movement and choice. Their accounts reinforce previous studies that have highlighted the relationship between leisure engagement and wellbeing [63], but also those that have problematized the relationship between leisure and freedom [12]. Overall, it is interesting to note that some refugees in Coffs Harbour have worked to redefine their leisure meanings, adapting or adopting new practices and experiences, while others struggle to settle between their cultural norms and performances to the new meanings of leisure they have been exposed to in their settlement countries. Again, there is a need for building a new vocabulary for leisure.

The last emergent theme highlighted the similarities experienced by humanitarian migrants and other vulnerable groups in society. Our results corroborate the relationship between socio-economic disadvantage and leisure participation [64] when the results from the modelling process showed that those who do not feel that cost is an important barrier for their participation in PA, engage in more PA. In addition to cost, language barriers, time and lack of knowledge of where to go featured as the main barriers for participation in leisure-time PA, a finding that corroborates recurrent results found in studies of leisure participation among disadvantaged populations [65–67]. In our study, however, only cost showed a significant relationship with levels of PA engagement. The lack of correlation with actual levels of PA engagement in these other categories, however, does not reduce their importance for the individuals who perceived these as barriers, as addressing them might further improve their engagement. Similar to our results on cost and on language barriers, a recent study with asylum seekers based in Perth found that these were among the major barriers for PA engagement among this group, together with mental health issues, which was consistently raised by our participants in interviews [51]. A study of refugee and other immigrants in the United States also found that lack of time and lack of local knowledge were important barriers to participation in PA [68], again supporting some of our findings.

Companionship was also seen to play an important role in leisure engagement, with those who indicated that companionship is not an important, or is only a somewhat important, barrier for participation engaged in less PA than those who indicate that companionship was very important–this suggests that group engagement may reinforce participation, which is, again, a recurrent finding in the literature [69]. Wieland and collaborators [68] discuss how refugee migrants were motivated to engage in PA by social support from family, friends and communities to be physically active, which reinforces our findings and suggest that efforts to increase

PA participation should consider social connections and development of social capital as part of the program.

Despite the insights provided by the material collected, it is important to also acknowledge this study's limitations. The main issue relevant to the findings reported here relate to the original aims of and approach to the study, where leisure engagement more broadly was not at the centre. As a consequence, the experiences of leisure explored were significantly focused on leisure-time physical activity, which we acknowledge as being restrictive. Given the challenges associated with the often acritical view of the relationship between health and physical activity found in public health policy [47], this focus can be viewed as yet another attempt to reproduce current discourses that disregard the myriad ways in which people engage (or not) with leisure practices. We hope, however, that our analysis of the data, including the close engagement with the community, has made it clear that this is but one of the leisure experiences with which refugees may engage and that are worth exploring.

It is also worth noting that focus groups, surveys and interviews were all conducted in English, with interpreters used only during survey completion. This clearly has an impact on the characteristics of participants and how much we have been able to learn from the diversity of experiences available in the community. This limitation notwithstanding, the findings presented here do provide valuable insights into the experiences of refugees in the region and more broadly, given the overall context of refugee resettlement in Australia and globally.

## Conclusions

In this article we have discussed how leisure studies scholars have repeatedly challenged the common narratives that reduce leisure experiences and pursuits to activities undertaken during one's "free time" and exposed the complexities involved in this simplistic view of leisure, particularly as it relates to the wellbeing of those experiencing disadvantage. Our quantitative results found that PA patterns prior to settlement, work status, and placing a high importance on cost, culture and companionship as barriers for participation were important predictors of levels of leisure-time PA. Qualitative results provided further insights into what leisure means in the lives of refugee migrants, challenging assumptions that leisure is perceived universally as beneficial and highlighting the need to consider the meanings associated with leisure more broadly by the refugee community if positive health outcomes are to be sought and achieved.

Although the refugee community in Coffs Harbour more specifically, and Australia more generally, present characteristics that are particular to their specific context, the discussion presented here may be clearly extrapolated to other refugee populations given the common, shared experience of displacement and difficulties in adapting to and integrating into a new culture and society. In this paper, we have argued that leisure is a powerful tool; one that can be used by program officers, policymakers and others who work closely with refugee communities across the world to better support their resettlement experience, particularly by supporting refugees to expand or regain access to their "vocabulary for recreation". This concept, a term used by one of our participants, provides a framework to explore refugee leisure experiences as it encapsulates the challenges experienced by humanitarian migrants to meaningfully engage in leisure. More than addressing the high level of psychological distress experienced by this population, or supporting their integration into a new culture and community, this approach addresses the human right to leisure and can be achieved through the universal language of sport, dance, games and recreation, but also through other non-physical leisure activities, such as arts and craft, among many others.

The intricacies associated with resettlement as well as the distinct mental health challenges experienced by refugee migrants lead to experiences of settlement that are clearly complex and

multifaceted. Bringing leisure to the forefront of discussions around service provision and policy making concerned with the wellbeing of the refugee migrant population, particularly focusing on support for the (re)development of a vocabulary for recreation, may well be a step forward in the journey to healing and belonging.

## Supporting information

**S1 File. Focus group schedule.**
(DOCX)

**S2 File. Survey.**
(PDF)

**S3 File. Interview schedule.**
(DOCX)

**S1 Table. Demographic characteristics.**
(DOCX)

**S2 Table. Health indicators.**
(DOCX)

## Acknowledgments

The authors would like to thank Sanctuary Australia Foundation for their invaluable support throughout this study and beyond. Their knowledge of the Coffs Harbour refugee community was instrumental in allowing this project to run successfully. We would also like to thank all participants for their generosity and invaluable contribution to this study.

## Author Contributions

**Conceptualization:** Arianne C. Reis, Sandro Sperandei.

**Data curation:** Arianne C. Reis, Matthew Bojanic, Sandro Sperandei.

**Formal analysis:** Arianne C. Reis, Sandro Sperandei.

**Investigation:** Arianne C. Reis, Komla Lokpo, Matthew Bojanic.

**Methodology:** Arianne C. Reis, Sandro Sperandei.

**Project administration:** Arianne C. Reis.

**Resources:** Arianne C. Reis.

**Software:** Arianne C. Reis, Sandro Sperandei.

**Supervision:** Arianne C. Reis.

**Validation:** Sandro Sperandei.

**Writing – original draft:** Arianne C. Reis.

**Writing – review & editing:** Arianne C. Reis, Sandro Sperandei.

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
