## [Decision Letter · Decision Letter 0]

1 Jul 2020

PONE-D-20-07177

In search of a “vocabulary for recreation”: Leisure-time physical activity among humanitarian migrants in regional Australia

PLOS ONE

Dear Dr. Reis,

Thank you for submitting your manuscript to PLOS ONE. After careful consideration, we feel that it has merit but does not fully meet PLOS ONE’s publication criteria as it currently stands. Therefore, we invite you to submit a revised version of the manuscript that addresses the points raised during the review process.

The reviewers raised several concerns concerning aspects of the methodology and asked for clarification with some of the terminology used in the manuscript. The full reviewer comments can be found below.

We look forward to receiving your revised manuscript.

Kind regards,

Natasha McDonald, PhD

Associate Editor

PLOS ONE

Journal Requirements:

Additional Editor Comments (if provided):

Reviewers' comments:

Reviewer's Responses to Questions

**Comments to the Author**

1. Is the manuscript technically sound, and do the data support the conclusions?

Reviewer #1: Yes

Reviewer #2: Yes

2. Has the statistical analysis been performed appropriately and rigorously? 

Reviewer #1: Yes

Reviewer #2: Yes

3. Have the authors made all data underlying the findings in their manuscript fully available?

Reviewer #1: No

Reviewer #2: Yes

4. Is the manuscript presented in an intelligible fashion and written in standard English?

Reviewer #1: Yes

Reviewer #2: Yes

5. Review Comments to the Author

Reviewer #1: The lack of public availability for the data has been explained by the authors as part of the submitted manuscript.

Please see full review below:

This paper addresses the meanings and experiences of leisure and their health implications for refugees in Australia's Coffs Coast. The research design and methodology of the paper are sound and in line with the journal's publication criteria. The data appropriately inform the discussion and conclusions, however a number of minor corrections are needed to clarify key elements/arguments of this study's context and design before this manuscript is accepted for publication. Please see below the list of corrections required:

At p. 3 lines 36-37, the authors claims that “In 2015, Australia received the third-largest number of refugees globally” while the cited data seem to refer only to the number of re-settled refugees. This is somehow very unclearly acknowledged at line 37, but this statement and the data mentioned would need much more clarity, as in fact UNHCR data (2016) showed that there were 21.3 million refugees globally in 2015. Such a clarification would be fundamental as the authors' claim about Australia's refugees reception seems to refer only to re-settled refugees, which only account for a very limited fraction of refugees globally. Secondly, clarifying that the numbers mentioned by the authors only refer to re-settled refugees, and not to refugees numbers overall, is important to avoid the unintended consequence of considering only re-settled refugees as “true” refugees worth considering in statistics.

While employing the term “leisure” in the article from the title, the authors seem to have focused instead in the data collection to the participants' physical activities. As such, not much is known about other leisure practices that could be less directly implying physical activity, but can be defined as leisure (e.g. music-making, drama/theatre participation, sport fandom, arts, etc.), and could still be relevant to the authors' main focus on health. More elaboration would thus be needed as to why a specific focus on PA was chosen by the authors, but the arguably wider definition of leisure was used (instead of simply physical activity). There is a passing comment on this at p. 18, lines 422-424, but more elaboration is needed, and ideally much earlier in the paper (e.g. while setting out the research design and aims).

Some contextual information would be relevant to add to address the specificities of the research context (e.g. to understand how such specificities played a role in influencing the participants leisure meanings and experiences), but also to shed more light on the research design process involved in the study. As such, some more elaboration on demographic, historical, and social domains of the context where the research was conducted would be needed. Furthermore, as the authors mention local “partners” more elaboration and contextual information would be important to explain how the collaboration started (e.g. how partners were involved in the project, who reached out to whom and why), what was the rationale for the collaboration across partners, how the community partners used (or aim to use) the research findings. This information could be relevant to provide important guidelines and references in terms of research design for scholars aiming to work with refugees and community organisations involved in supporting/advocating for refugees.

A list of minor, but still relevant passages that would need clarification/amendments:

1. Sentence at p. 3 lines 40-46 too lengthy and unclear. Recommended to split it in two clearer and shorter sentences.

2. P. 6 Line 128 Why “mixed-methods research is a challenging area within which to work”? Clarify/elaborate or delete?

3. p. 8 line 133 “themes” possibly better wording than “names”?

4. P. 12 line 270, participant presented as “female from Sierra Leone”, while at line 273, same page, presented as “female from Liberia”

5. P. 16, lines 380-381, “when you settle” could be worded better using the term “when refugees settle”, etc.

6. p. 17, lines 409-410 “our quantitative study did not find gender-based differences in participation, suggesting that other factors might be at play in this community” can the authors elaborate more on these “other factors” that might be at play, or indicate more clearly this as a further line of enquiry for research on the topic?

7. P. 20 line 473 “PA”

8. P. 21 lines, 492-494 the authors state “Such a position is only recurrent in their narratives because it is equally entrenched in public discourse.” to what extent the data available justify this direct correlation between public discourses on leisure and the participants' narratives on leisure engagement? To what extent instead issues such as poverty, the “cold-panic” and ongoing worries for family and relationships elsewhere, and further factors play a role in making leisure to seem “impossible” for some of the participants? As these factors are considered in the paper's findings and discussion, it is likely just a matter for the authors to re-word the sentence above and allow for more openness in identifying factors at stake in what emerged from the data, consistently with their discussion overall.

Reviewer #2: This mixed-methods study explored the meanings and practices of leisure found in the experiences of day-to-day living of refugee migrants resettled in a regional Australian town. I read the paper with great interest. Overall I found the paper informative and well-designed. I hope the authors find the comments helpful.

1. Shorten the part of “Leisure Meanings” and summary key findings from the literature review at the end.

2. Give an overall description of the study design at the beginning of the method. How were the three stages of the study linked? Specify the (different) purposes of two qualitative studies (focus-group and in-depth interview) before and after the quantitative study.

3. Kindly provide the interview guidelines for both the first and third studies, at least some key questions to help the reader understand the content. It might be better to provide backgrounds of the interviewers and interviewees as well.

4. Any pilot testing of the interview guidelines and survey questionnaire?

5. Any information could be provided in terms of quality control (both the quantitative and qualitative parts)?

6. For each quote, please provide the participant’s age, gender, and origin at the end in a bracket to provide supplemental information.

7. I noticed that (Line 215) most participants had no previous schooling (28.9%), although the majority were currently studying (51.9%). Is this low literacy affect the participants’ understanding of the study, particularly the second part of the survey that used validated scales? For example, SF-36 is quite long and complex. How was the questionnaire administrated? Was the questionnaire used in their original language?

8. Line 233, I totally agree, “…further quantitative results are presented and discussed together with qualitative ones in order to facilitate integration of the knowledge gained from all phases of the study”. I have a few suggestions for the result presenting in this section:

a) The current three themes were a bit general. Some keywords in these themes (“It Makes You Lazy”, A similar story”) were not specified. Details should be given, such as a similar story to what?

b) Is that possible to summarize a few subthemes under each theme? For example, In this context, participation in PA was seen as superfluous and a privilege they could not afford, both in terms of finance, time and also motivation. Here “finance”, “time”, and “motivation” and good subthemes to present.

c) Some sentences in the Results should be moved to the Discussion. For example, (Line 381) some of them are not too dissimilar to those experienced by other marginalised groups in our (Western) societies. Please present results derived from the present study only, and move potential explanations and comparisons (with others) to the Discussion.

d) I did not see a clean message from the quantitative part; some key sentences would be useful.

e) How about the validity of the SF-36 and K10 in the present study?

f) It might be better to move the two supplementary tables to the main paper because they provide essential information.

9. Discuss a bit about the representativeness of the sample. Line 205 …”this sample represents approximately one third of the entire population”, this does not necessarily mean good representativeness. Any issues raised because of changing from respondent-driven sampling to convenience sampling in the second stage?

10. Based on the findings presented in this study, what implications could be made to program officers or policymakers?

11. How could results derived from the present sample be generalized to other migrate populations worldwide?

12. Discuss the limitations of the present study.

6. PLOS authors have the option to publish the peer review history of their article (what does this mean?). If published, this will include your full peer review and any attached files.

Reviewer #1: **Yes: **Dr. Nicola De Martini Ugolotti

Reviewer #2: No

---

## [Author Response · Author response to Decision Letter 0]

31 Jul 2020

Thanks for the valuable and constructive feedback provided. We have responded to all comments provided by the reviewers in the attached "Response to Reviewers" file that explains in detail how each of the comments were addressed in the manuscript. We feel that the manuscript has become much stronger as a result.

---

## [Decision Letter · Decision Letter 1]

17 Aug 2020

PONE-D-20-07177R1

In search of a “vocabulary for recreation”: Leisure-time physical activity among humanitarian migrants in regional Australia

PLOS ONE

Dear Dr. Reis,

Thank you for submitting your manuscript to PLOS ONE. After careful consideration, we feel that it has merit but does not fully meet PLOS ONE’s publication criteria as it currently stands. Therefore, we invite you to submit a revised version of the manuscript that addresses the points raised during the review process.

The reviewer for the manuscript feels that you have addressed the previous concerns, but feels that there is one more 'minor but substantial' issue that should be addressed before publication. 

We look forward to receiving your revised manuscript.

Kind regards,

Natasha McDonald, PhD

Associate Editor

PLOS ONE

Journal Requirements:

Additional Editor Comments (if provided):

Reviewers' comments:

Reviewer's Responses to Questions

**Comments to the Author**

1. If the authors have adequately addressed your comments raised in a previous round of review and you feel that this manuscript is now acceptable for publication, you may indicate that here to bypass the “Comments to the Author” section, enter your conflict of interest statement in the “Confidential to Editor” section, and submit your "Accept" recommendation.

Reviewer #1: (No Response)

2. Is the manuscript technically sound, and do the data support the conclusions?

Reviewer #1: Yes

3. Has the statistical analysis been performed appropriately and rigorously? 

Reviewer #1: Yes

4. Have the authors made all data underlying the findings in their manuscript fully available?

Reviewer #1: Yes

5. Is the manuscript presented in an intelligible fashion and written in standard English?

Reviewer #1: Yes

6. Review Comments to the Author

Reviewer #1: The authors have addressed all the comments made in the previous review, and as a result the paper now offers a valid contribution to the topic it addresses.

I would only recommend one final, minor, yet substantial revision before recommending this paper for publication.

This relates to the automatic association of being a refugee with trauma that is often made in several points of the paper. Important contributions in the field of refugee studies have for some time challenged this automatic association, by convincingly arguing that, while it would be foolish to claim that displacement does not cause distress of many kinds, we cannot consider trauma as an axiom in relation to refugees' lives, as the implication of this association is to unwittingly expect these characteristics in this population (see Malkki, 1995, 1996; Fassin, 2005). These considerations point to avoid addressing refugees as “kind of persons” with shared and generalisable experiences, but as a broad, descriptive rubric that includes within it different socio-economic statuses, personal histories and psychological situations (see Malkki, 1995, p. 496). Despite being mentioned once by research participants in the data, “trauma” seems to have been used in the article a bit as a “catch-all” term in the paper and automatically associated with being a refugee. In other parts of the article, participants' experiences are qualified as traumatic despite seemingly addressing emotional and psychological experiences (e.g. ties and concerns for family members described as “baggage”) that would be misplaced to define as “trauma”. I would recommend caution in automatically associating trauma as an intrinsic qualifier of “being a refugee”, as these kind of automatic associations perpetuate assumptions and expectations of what a “real” refugee is that have some significant implications when these inform practice and policy. This is relevant as in the conclusions the authors discuss the generalisability of their findings exactly in terms of the “common experiences of trauma, etc.” (p. 28 line, 673 in the manuscript) can unwittingly contribute to normalise such problematic framings. While I understand this addition addressed one of Reviewer 2's comments, I would recommend shifting the core of this statement. From the present statement that argues generalisability despite the specificity of the research context, the authors could shift to one that argues generalisability because of shared challenges that refugee populations face in contexts of re-settlement (e.g. under-employment, limited social opportunities and mental health pressures, etc.) that would be important to understand in light of refugees' populations diversity and complexity. This relatively minor change would make a substantial difference in the level of nuance and contribution of this valid study to highlighting the relevance of leisure domains in contexts' of refugees' resettlement.

7. PLOS authors have the option to publish the peer review history of their article (what does this mean?). If published, this will include your full peer review and any attached files.

Reviewer #1: **Yes: **Nicola De Martini Ugolotti

---

## [Author Response · Author response to Decision Letter 1]

20 Aug 2020

We have endeavoured to address the matters raised by the reviewer and have included a detailed response in the attached cover letter.

---

## [Editor Report · Decision Letter 2]

7 Sep 2020

PONE-D-20-07177R2

In search of a “vocabulary for recreation”: Leisure-time physical activity among humanitarian migrants in regional Australia

PLOS ONE

Dear Dr. Reis,

Thank you for submitting your manuscript to PLOS ONE. After careful consideration, we feel that it has merit but does not fully meet PLOS ONE’s publication criteria as it currently stands. Therefore, we invite you to submit a revised version of the manuscript that addresses the points raised during the review process.

Please see my comments below and provide a response. 

We look forward to receiving your revised manuscript.

Kind regards,

Andrew Soundy

Academic Editor

PLOS ONE

Additional Editor Comments (if provided):

I have been asked to act as a new editor. I have some questions about the methodology before I send it out to reviewers for a final check.

Points relating to methods

Study design

Needs to be framed within a framework like (happy for another to be used) CARE https://www.care-statement.org/checklist

Or GRAMMS if mixed methods

https://www.researchgate.net/profile/Roslyn_Cameron3/publication/299738288_Cameron_R_Dwyer_T_Richardson_S_Ahmed_E_and_Sukumaran_A_2013_'Lessons_from_the_field_Applying_the_Good_Reporting_of_A_Mixed_Methods_Study_GRAMMS_Framework'_Electronic_Journal_of_Business_Research_Metho/links/5704aca908aef745f71494eb/Cameron-R-Dwyer-T-Richardson-S-Ahmed-E-and-Sukumaran-A-2013-Lessons-from-the-field-Applying-the-Good-Reporting-of-A-Mixed-Methods-Study-GRAMMS-Framework-Electronic-Journal-of-Business-Researc.pdf

line 143 you state: “The methodology utilises positivist epistemology, a case study of Coffs Harbour, and constructivist theory-building”

A positivist epistemology is Representational - People can know the reality and objectively explain it. Constructivism is an epistemology based on one’s own experiences built up through learning within a group or culture it is a subjectivist epistemology. – the statement could be confusing for the reader so please qualify or give context to stages

Line 165-167 seems like results – can it be moved please.

Line 171 – you consider stages and it seems like a mixed methods study so needs a name up front and given a type.

You need to justify your sample size across stages, e.g., you talk about saturation please give a type (data or theoretical) of saturation you refer to. Identify when saturation was identified in what number of interview? You need to give your power calculation for the survey.

You need to justify eligibility criteria for each stage

You need to give details of the development of the outcome measures used or interview schedules including piloting, cognitive interviews, development stages.

Audit trail to document qualitative analysis needs to be given

You must talk about the level of integration of data - fully or partial - if it is mixed methods? if there is no integration then its multi methods?

---

## [Author Response · Author response to Decision Letter 2]

11 Sep 2020

We thank you for the additional feedback provided. We have responded to your requests in the attached "Response to Reviewers" file. We have also included a revised manuscript with tracked changes so that you can easily identify the changes made. 

Sincerely,

The authors

---

## [Editor Report · Decision Letter 3]

14 Sep 2020

In search of a “vocabulary for recreation”: Leisure-time physical activity among humanitarian migrants in regional Australia

PONE-D-20-07177R3

Dear Dr. Reis,

We’re pleased to inform you that your manuscript has been judged scientifically suitable for publication and will be formally accepted for publication once it meets all outstanding technical requirements.

Kind regards,

Andrew Soundy

Academic Editor

PLOS ONE
---

## [Editor Report · Acceptance letter]

23 Sep 2020

PONE-D-20-07177R3 

In search of a “vocabulary for recreation”: Leisure-time physical activity among humanitarian migrants in regional Australia 

Dear Dr. Reis:

I'm pleased to inform you that your manuscript has been deemed suitable for publication in PLOS ONE. Congratulations! Your manuscript is now with our production department. 

Kind regards, 

on behalf of

Dr. Andrew Soundy 

Academic Editor

PLOS ONE